# From Blind Spots to Gains:
# Diagnostic-Driven Iterative Training for Large Multimodal Models

Hongrui Jia [1*]   Chaoya Jiang [2* †]   Yongrui Heng [1]   Shikun Zhang [1]   Wei Ye [1†]

## Abstract

As Large Multimodal Models (LMMs) scale up and reinforcement learning (RL) methods mature, LMMs have made notable progress in complex reasoning and decision making. Yet training still relies on static data and fixed recipes, making it difficult to diagnose capability blind spots or provide dynamic, targeted reinforcement. Motivated by findings that test driven error exposure and feedback based correction outperform repetitive practice, we propose Diagnostic-driven Progressive Evolution (DPE), a spiral loop where diagnosis steers data generation and reinforcement, and each iteration re-diagnoses the updated model to drive the next round of targeted improvement. DPE has two key components. First, multiple agents annotate and quality control massive unlabeled multimodal data, using tools such as web search and image editing to produce diverse, realistic samples. Second, DPE attributes failures to specific weaknesses, dynamically adjusts the data mixture, and guides agents to generate weakness focused data for targeted reinforcement. Experiments on Qwen3-VL-8B-Instruct and Qwen2.5-VL-7B-Instruct show stable, continual gains across eleven benchmarks, indicating DPE as a scalable paradigm for continual LMM training under open task distributions. Our code, models, and data are publicly available at https://github.com/hongruijia/DPE.

## 1. Introduction

In recent years, as reinforcement learning methods (Guo et al., 2025; Zheng et al., 2025; Yu et al., 2025; Zhao et al., 2025; Gao et al., 2025) have matured, the reasoning ca- pabilities of Large Multimodal Models (LMMs) have im- proved substantially. Models such as GPT-5.2 (OpenAI Team, 2025), Claude Sonnet 4.5 (Anthropic Team, 2025), and Qwen3-VL (Bai et al., 2025a) show particularly strong performance on complex reasoning tasks. However, anno- tated data for multimodal reasoning remains scarce, making it difficult to support large scale training of LMMs(Liu et al., 2025a).

To mitigate this, prior works (He et al., 2025; Chen et al., 2025a; Liu et al., 2024; Thawakar et al., 2025; Sunil et al., 2026; Liu et al., 2025b) have proposed *self-evolving train- ing frameworks characterized by an iterative cycle of self- questioning and self-answering to continuously refine the model*. While these approaches have garnered significant at- tention, current methodologies are constrained by two funda- mental limitations: **1. Lack of Interpretable Diagnostics.** Driven by heuristic signals (e.g., perplexity) rather than ex- plicit failure attribution, existing methods lack a principled capability decomposition. Consequently, the evolutionary process pursues superficial complexity instead of addressing genuine capability gaps, resulting in unstable data quality and noise. **2. Scarcity of Visual Diversity.** Reliance on static image sets inherently restricts the semantic scope of training. While textual queries evolve, the immutable visual context limits the coverage of long-tail scenarios, causing performance on rare or complex concepts to plateau or even regress.

Research in educational psychology (Black & Wiliam, 1998; Hattie & Timperley, 2007) reveals that diagnosis and tar- geted correction are the pivotal determinants of learning efficiency. Inspired by this *"diagnose-and-correct"* mech- anism in human cognition, we propose **Diagnostic-driven Progressive Evolution (DPE)**. Mirroring these principles, DPE eschews indiscriminate data expansion. Instead, it pri- oritizes the diagnosis of capability gaps to steer targeted data generation and mixture optimization, effectively breaking the multimodal long-tail bottleneck.

Specifically, DPE consists of two key mechanisms: (1)Adap- tive Diagnosis. Before generating new data, a diagnostic agent analyzes the model's failure patterns to identify spe- cific weaknesses and capability blind spots. These insights are used to dynamically optimize the training data mixture,

---
[*]Equal contribution [1]Peking University [2]Shandong University. Correspondence to: Chaoya Jiang <jcy@sdu.edu.cn>, Wei Ye <wye@pku.edu.cn>.

*Proceedings of the 43rd International Conference on Machine Learning*, Seoul, South Korea. PMLR 306, 2026. Copyright 2026 by the author(s).

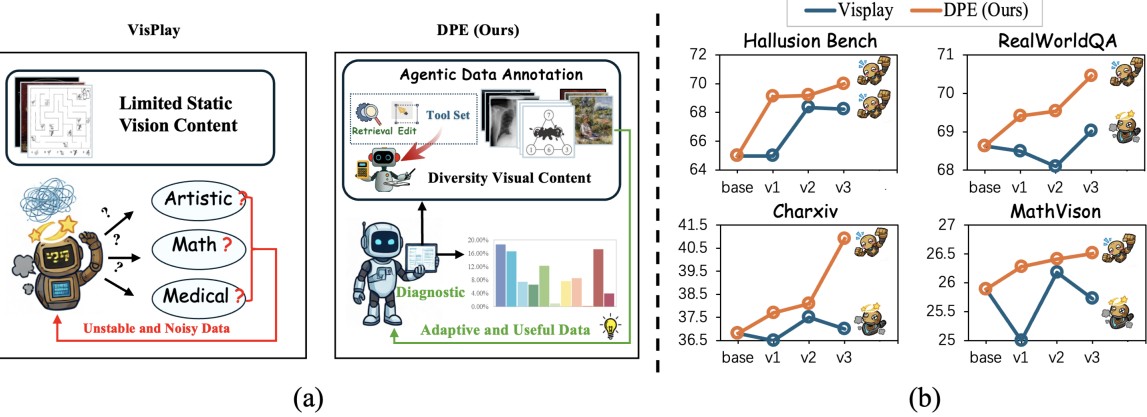

*Figure 1.* Due to the lack of interpretable diagnostics and scarcity of visual diversity, previous self-evolution frameworks can alleviate hallucination to some extent but fail to provide meaningful improvements on long-tail tasks such as mathematics and OCR. As a result, the model often exhibits instability or even degradation in these capabilities during the evolution process. In contrast, our DPE framework effectively addresses these blind spots and supports a more comprehensive and balanced progression of the model's abilities.

driving a closed loop of diagnosis, generation, and reinforcement for targeted capability improvement. (2) Tool-Use Data Evolution. Instead of relying on static datasets or template-based text rewriting, DPE employs a multi-agent system equipped with image search and editing tools. These agents collaboratively source and annotate diverse visual content from external pools, allowing the framework to construct high-quality, weakness-focused training samples with deliberately controlled distributions. Crucially, DPE is not constrained by a static dataset. It can adaptively source images from large external pools and construct targeted questions to form training data with deliberately controlled mixtures for reinforcement.

We apply our DPE framework to multiple models, including Qwen2.5-VL-7B-Instruct (Bai et al., 2025b) and Qwen3-VL-8B-Instruct (Bai et al., 2025a), and evaluate it on 11 challenging benchmarks that probe different aspects of multimodal reasoning, including MMMU (Yue et al., 2024b), CharXiv (Wang et al., 2024b), and MathVision (Wang et al., 2024a). Experiments show that, compared with static data training methods, our framework can improve model capabilities broadly using only a small amount of training data. Further analysis indicates that, as training iterates, the diagnosis mechanism noticeably improves training stability, while the unlabeled multimodal annotation mechanism effectively alleviates bottlenecks caused by static data. Our main contributions are as follows:

- We propose a novel Diagnostic-driven Progressive Evolution (DPE) training paradigm that targets model blind spots through a diagnosis, generation, and reinforcement loop, mitigating diminishing marginal returns during training and avoiding long tail coverage issues induced by static data.

- We demonstrate the efficiency of DPE on multiple open

source models. With only 1000 training examples, it achieves broad improvements in multimodal reasoning.

- We provide systematic analyses that quantitatively evaluate how the diagnosis mechanism affects training stability, offering a new direction for addressing long tail challenges in improving multimodal reasoning.

## 2. Related Work

### 2.1. Reasoning with Large Multimodal Models

The success of reinforcement learning (RL) in enhancing the reasoning capabilities of Large Language Models (LLMs) (Guo et al., 2025; Zheng et al., 2025; Zhao et al., 2025) has spurred similar advancements in Large Multimodal Models (LMMs). Recent works focus on establishing verifiable reward mechanisms to align visual reasoning. For instance, VLM-R1 (Shen et al., 2025) and RRVF (Chen et al., 2025b) introduce rule-based and rendering-based feedback loops, respectively, to ground reasoning in verifiable signals. Others, such as Vision-SR1 (Li et al., 2025b) and SRPO (Wan et al., 2025), leverage self-consistency and self-reflection to mitigate hallucinations and refine reasoning trajectories. OVR (Wei et al., 2025) further explores cold-start strategies to transfer cognitive behaviors from language to vision. Despite these strides, most RL-based LMMs rely heavily on static datasets or expensive annotations. They often lack the mechanism to dynamically adapt the data distribution to the model's evolving capabilities, leading to inefficiencies where models over-train on mastered samples while neglecting long-tail weaknesses.

### 2.2. Self-Evolving Multimodal Frameworks

To address data scarcity, self-evolving paradigms (He et al., 2025; Chen et al., 2025a; Liu et al., 2024) have emerged,

where models improve via self-generated feedback. Existing approaches can be broadly categorized into *filtering-based* and *generative* methods. Filtering strategies, such as M-STAR (Liu et al., 2024) and EvoLMM (Thawakar et al., 2025), utilize uncertainty metrics (e.g., entropy) or process reward models to select high-quality samples from noisy generations. Generative frameworks, exemplified by VisPlay (He et al., 2025) and IREASONER (Sunil et al., 2026), employ a proposer-solver loop where a generator creates new queries and a solver verifies them using consistency checks. More recent agentic approaches (Liu et al., 2025b; Pan et al., 2025) incorporate tool-use and multi-agent collaboration to enhance the reliability of self-evaluation. However, a critical limitation remains: current self-evolving pipelines typically operate in a "blind" manner. They generate or filter data based on general quality metrics rather than explicitly *diagnosing* the model's specific failure modes. This often results in distribution drift or mode collapse during iterations, as the generated data fails to target the model's actual cognitive blind spots.

## 3. Methods

### 3.1. Overall Framework

As shown in Figure 2, we propose **Diagnostic-driven Progressive Evolution (DPE)**, a closed-loop training framework that steadily improves large multimodal models (LMMs) under scarce multimodal supervision and long-tail coverage gaps. Different from prior self-evolution methods that depend on static image sets and heuristic signals, DPE iteratively performs **diagnosis**, **targeted generation**, and **reinforcement-based updating**. In each iteration, DPE explicitly controls both the training-data category composition and question emphasis, aligning training resources with current capability blind spots and reducing instability and diminishing returns on long-tail skills.

Let the policy at iteration $k$ be $\pi_{\theta^{(k)}}$. DPE constructs a training set $\mathcal{T}^{(k)}$ and updates parameters to $\theta^{(k+1)}$ via reinforcement learning with verifiable rewards:

$$\theta^{(k+1)} = \mathcal{A}_{\text{RL}}\big(\theta^{(k)}; \mathcal{T}^{(k)}\big), \mathcal{T}^{(k)} = \mathcal{A}_{\text{gen}}\big(\mathcal{R}^{(k)}\big), \mathcal{R}^{(k)} = \mathcal{A}_{\text{diag}}\big(\pi_{\theta^{(k)}}\big), \quad (1)$$

where $\mathcal{A}_{\text{diag}}$, $\mathcal{A}_{\text{gen}}$, and $\mathcal{A}_{\text{RL}}$ are the diagnosis, generation, and RL-update operators, respectively, and $\mathcal{R}^{(k)}$ is a structured diagnostic report.

### 3.2. Diagnostic Mechanism

The diagnostic mechanism provides an **interpretable and actionable** assessment of the current policy $\pi_{\theta^{(k)}}$ at the start of each iteration, and converts it into constraints/instructions for the next-round data generation. Instead of heuristic proxies (e.g., perplexity or reward averages), our diagnosis performs **explicit failure attribution** and **capability**

**decomposition**: it identifies where the model fails, which capability dimension is responsible, and recurring error patterns, enabling stable targeted evolution.

We map multimodal logical reasoning into a capability space $\mathcal{C} = \{c_1, c_2, \ldots, c_K\}$ with $K = 12$ dimensions, including geometry images, medical images, statistical charts, text-intensive images, flow diagrams, mathematical formulas, spatial maps, natural scenes, daily objects, artworks, architectural images, and others.

**Diagnostic sampling and step-aware scoring.** At iteration $k$, we sample $N = 200$ instances from a diagnostic pool $\mathcal{D}_{\text{diag}}$:

$$\{(I_n, q_n, a_n, c_n)\}_{n=1}^N \sim \mathcal{D}_{\text{diag}}. \quad (2)$$

The model produces $\hat{y}_n \sim \pi_{\theta^{(k)}}(\cdot \mid I_n, q_n)$, which is scored by diagnostic agents:

$$z_n = v(\hat{y}_n, a_n), \quad (3)$$

where $v(\cdot)$ evaluates both reasoning steps and final results; we convert $z_n$ into a scalar correctness signal for aggregation. For each category $c$, we compute counts and accuracy:

$$N_c = \sum_{n=1}^N \mathbb{I}[c_n = c], \qquad \text{Acc}_c = \frac{1}{N_c} \sum_{n=1}^N \mathbb{I}[c_n = c] \cdot z_n.$$

**Failure attribution and diagnostic summary.** Beyond category accuracy, agents analyze the error set $\mathcal{E}_c = \{n \mid c_n = c, z_n = 0\}$ and summarize recurring patterns as $\mathcal{F}_c$ (e.g., OCR: missing lines/misaligned regions; charts: ignored axis units/legend mismatch; math: dropped steps/symbol parsing errors; multi-image: entity misalignment/incorrect reference resolution). These attributions are injected into generation as executable prompts to control focus and difficulty.

**From diagnosis to category proportions.** A key output is the category proportion vector $\boldsymbol{\alpha}^{(k)}$ for next-round generation. We assign unnormalized weights $\tilde{\alpha}_c$ according to segmented ranges of $\text{Acc}_c$, and normalize:

$$\alpha_c^{(k)} = \frac{\tilde{\alpha}_c}{\sum_{c'=1}^C \tilde{\alpha}_{c'}}. \quad (4)$$

**Structured diagnostic report.** The final report is

$$\mathcal{R}^{(k)} = \Big(\boldsymbol{\alpha}^{(k)}, \{\mathcal{F}_c^{(k)}\}_{c=1}^C, \{\mathcal{H}_c^{(k)}\}_{c=1}^C\Big), \quad (5)$$

where $\boldsymbol{\alpha}^{(k)}$ controls category quotas, $\mathcal{F}_c^{(k)}$ records within-category weaknesses, and $\mathcal{H}_c^{(k)}$ provides actionable generation instructions (e.g., stronger localization, longer reasoning chains, stricter answer formats).

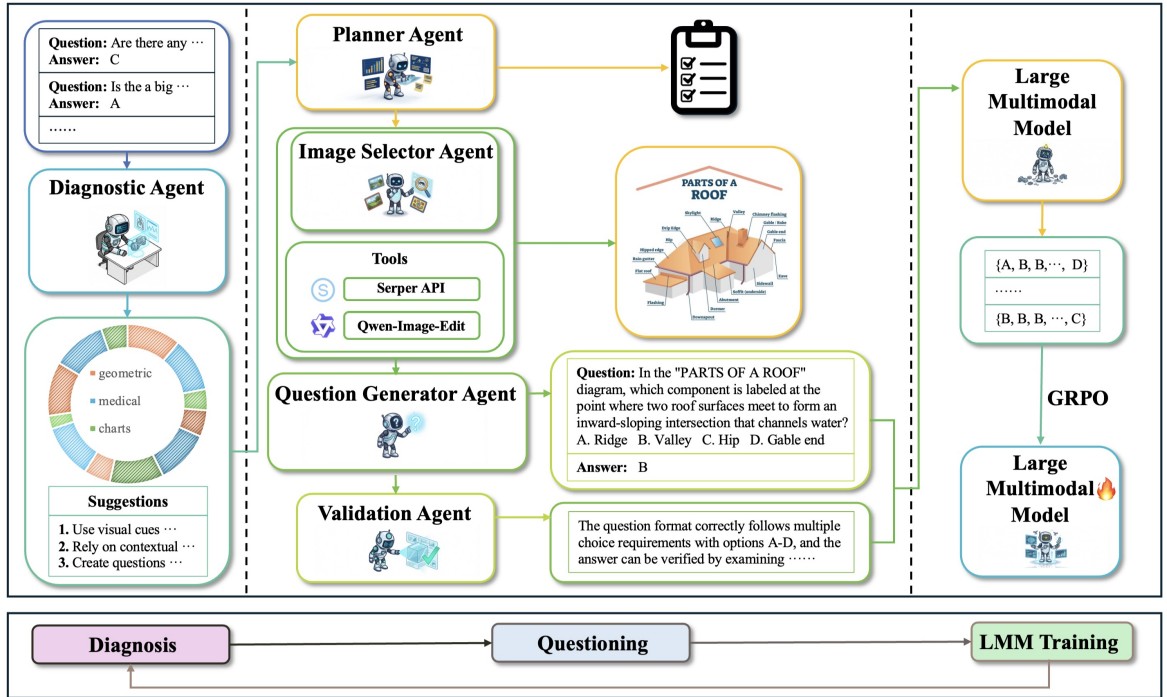

*Figure 2.* Overview of the DPE framework.

## 3.3. Multiple Agents Questioner System

The Multiple Agents Questioner System converts the diagnostic report $\mathcal{R}^{(k)}$ into a collection of training samples with **controllable distribution**, **controllable quality**, and **verifiable answers**, which are then used for RLVR optimization (e.g., GRPO). Unlike self-evolution methods that only rewrite text instructions over a static image set, our system coordinates four specialized agents for planning, retrieval and editing, question construction, and validation. It explicitly enforces both **category quota constraints** and **quality gating constraints**, thereby improving the relevance of the generated data while maintaining training stability.

**Category quota constraint and dataset formalization.** Let the diagnostic mechanism output the category proportions at iteration $k$ as $\boldsymbol{\alpha}^{(k)} = (\alpha_1^{(k)}, \ldots, \alpha_C^{(k)})$. Given a target generation budget of $M$ samples for this iteration, the category quota is defined as $m_c = \left\lfloor M \cdot \alpha_c^{(k)} \right\rfloor, c = 1, \ldots, C$, and the final generated set $\mathcal{T}^{(k)}$ is required to satisfy the hard constraint

$$\sum_{(I,q,a,c)\in\mathcal{T}^{(k)}} \mathbb{I}[c = c'] = m_{c'}, \quad \forall c' \in \{1, \ldots, C\} \quad (6)$$

The system outputs a training dataset $\mathcal{T}^{(k)} = \{(I_j, q_j, a_j, c_j)\}_{j=1}^{M}$, where $I_j$ can be a single image or a multi-image composition, and the reference answer $a_j$ must be verifiable to support stable reinforcement learning with verifiable rewards.

As shown in Figure 2, the system consists of four modules: **Planner Agent**, **Image Selector Agent**, **Question Generator Agent**, and **Validation Agent**. They collaborate through the shared quota state $\mathbf{n}$ and the diagnostic information $\mathcal{R}^{(k)}$, forming a pipeline that covers plan construction, data instantiation, and quality verification.

**Planner Agent.** The Planner agent translates diagnostic outputs into executable instructions at the level of individual samples. For the $j$-th sample to be generated, the Planner outputs:

$$\text{plan}_j = \left(c_j, \text{ req}_j^I, \text{ req}_j^Q, \text{ dir}_j\right), \quad (7)$$

where $c_j$ is the target category and must satisfy the quota constraint $n_{c_j} < m_{c_j}$, $\text{req}_j^I$ specifies image requirements (e.g., containing readable text regions, axes and legends in a chart, or two comparable views sharing the same entities), $\text{req}_j^Q$ specifies question requirements (e.g., multiple-choice versus numeric, whether a unit is required, and whether a structured output format is required), and $\text{dir}_j$ specifies a direction constraint that targets the model's weaknesses within this category, derived from $\mathcal{F}_{c_j}^{(k)}$ and $\mathcal{H}_{c_j}^{(k)}$.

**Image Selector Agent.** Given $\text{req}_j^I$, the Image Selector agent obtains the visual input from an external image pool $\mathcal{P}_{\text{ext}}$ and performs editing or composition when necessary:

$$I_j = \phi(\mathcal{P}_{\text{ext}}, \text{ req}_j^I), \quad (8)$$

where $\phi(\cdot)$ represents a pipeline that includes retrieval, candidate filtering, editing or fusion, and final selection. The

agent provides three types of capabilities: (1) **Search**, which performs large-scale retrieval using keywords, category tags, and structural hints (e.g., "bar chart with legend"); (2) **Filter**, which applies basic quality screening to candidates, including resolution thresholds, image size, consistency between image type and plan requirements (e.g., confirming chart structure or sufficient text regions), and basic readability checks; (3) **Edit/Compose**, which constructs targeted scenarios via cropping, overlaying text, stitching multiple images, or fusing local regions, especially for long-tail concept coverage and boundary-case construction. This tool-use design frees DPE from the limitations of static datasets, expands semantic coverage at the image level, and enables rapid and targeted reproduction of blind spots identified during diagnosis.

**Question Generator Agent.** Given the image $I_j$ and the planning instructions, the Question Generator agent produces a question $q_j$ and a reference answer $a_j$:

$$(q_j, a_j) = \psi\big(I_j,\ \mathrm{req}_j^Q,\ \mathcal{H}_{c_j}^{(k)}\big) \tag{9}$$

In practice, this stage strictly follows the Planner's global plan. When the Planner detects that a category quota has been reached, namely $n_c = m_c$, it skips that category and proceeds to plan the next sample, ensuring that the final data distribution matches the proportions specified by the diagnostic mechanism. Meanwhile, the Planner refines image requirements for the Image Selector agent and specifies clear question directions for the Question Generator agent based on category-level focus and difficulty recommendations, forming a jointly consistent plan between visual constraints and question directions.

**Validation Agent.** Since self-generated samples may suffer from category drift, underspecification, or answer inconsistency, we introduce a Validation agent to explicitly gate data quality and ensure that accepted samples meet minimum standards. For a candidate sample $s_j = (I_j, q_j, a_j, c_j)$ with plan $\mathrm{plan}_j$, we define the following checks: category consistency $g_{\mathrm{cat}}(s_j, \mathrm{plan}_j)$, solvability and information completeness $g_{\mathrm{sol}}(s_j)$, answer verifiability $g_{\mathrm{ver}}(s_j)$, and format compliance $g_{\mathrm{fmt}}(s_j, \mathrm{req}_j^A)$. The final acceptance condition is

$$g(s_j) = g_{\mathrm{cat}} \cdot g_{\mathrm{sol}} \cdot g_{\mathrm{ver}} \cdot g_{\mathrm{fmt}} \tag{10}$$

If $g(s_j) = 1$, the sample is added to the training set and $\mathbf{n}$ is updated. Otherwise, the sample is discarded and the system re-generates a new candidate. This gating procedure reduces training noise, improves the stability of RLVR optimization, and prevents miscategorized samples from corrupting quota statistics and causing distribution drift.

## 3.4. LMM Training

We optimize the target multimodal model using GRPO. For each prompt $x$, the old policy $\pi_{\theta_{\mathrm{old}}}$ generates $G$ trajectories $y_i = (o_{i,1}, \ldots, o_{i,|y_i|}) \sim \pi_{\theta_{\mathrm{old}}}(\cdot \mid x)$, $i = 1, \ldots, G$, where $o_{i,t}$ denotes the $t$-th token in the $i$-th trajectory. Each trajectory is assigned a scalar reward $r_i = r(x, y_i) \in \mathbb{R}$.

GRPO optimizes the following clipped surrogate objective:

$$J_{\mathrm{GRPO}}(\theta) = \mathbb{E}_{x \sim \mathcal{D},\ \{y_i\} \sim \pi_{\theta_{\mathrm{old}}}} \left[ \frac{1}{G} \sum_{i=1}^{G} \frac{1}{|y_i|} \sum_{t=1}^{|y_i|} \min\Big(\rho_{i,t} A_{i,t}, \right.$$
$$\left. \mathrm{clip}(\rho_{i,t}, 1-\varepsilon, 1+\varepsilon)\, A_{i,t}\Big)\ -\ \beta\, \mathrm{KL}(\pi_\theta \,\|\, \pi_{\mathrm{init}}) \right] \tag{11}$$

where $\rho_{i,t} = \frac{\pi_\theta(o_{i,t}|x, o_{i,<t})}{\pi_{\theta_{\mathrm{old}}}(o_{i,t}|x, o_{i,<t})}$, $\varepsilon$ is the PPO-style clipping threshold, $\beta > 0$ controls the strength of KL regularization, $\pi_{\mathrm{init}}$ is a reference policy, and $\pi_\theta$ is the current trainable policy.

**Group-normalized advantages.** A key design of GRPO is the trajectory-level *group-normalized advantage*:

$$\hat{A}_i = \frac{r_i - \mathrm{mean}(r_1, \ldots, r_G)}{\mathrm{std}(r_1, \ldots, r_G)}. \tag{12}$$

**A maximum-entropy view of learnability.** From the perspective of maximum-entropy policy improvement, given a reward function $r(x, y)$ with $x = (I, q)$, the optimal policy satisfies

$$\pi^*(y \mid x) \propto \pi_{\mathrm{init}}(y \mid x) \exp(r(x, y)/\beta) \tag{13}$$

and the reverse KL divergence admits the expression

$$\mathrm{KL}(\pi_{\mathrm{init}} \,\|\, \pi^*) = \frac{1}{\beta}\Big(V^*(x) - \mathbb{E}_{\pi_{\mathrm{init}}}[r(x, y)]\Big) \tag{14}$$

Following prior work (Bae et al., 2025; Shi et al., 2025; Bu et al., 2025; Huang et al., 2025), the associated soft value function is

$$V^*(x) = \beta \log \mathbb{E}_{y \sim \pi_{\mathrm{init}}}\left[\exp(r(x, y)/\beta)\right] \tag{15}$$

For binary rewards $r \in \{0, 1\}$, let the pass rate be

$$p(x) = \mathbb{E}_{\pi_{\mathrm{init}}}[r(x, y)]. \tag{16}$$

Then

$$V^*(x) = \beta \log\Big((1 - p(x)) + p(x) \exp(1/\beta)\Big), \tag{17}$$

and a second-order lower bound yields

$$\mathrm{KL}(\pi_{\mathrm{init}} \,\|\, \pi^*) \geq \frac{p(x)\big(1 - p(x)\big)}{2\beta^2}. \tag{18}$$

This bound depends only on the variance term $p(x)(1 - p(x))$. It vanishes when $p$ is close to 0 or 1, and it is maximized near $p = 0.5$. Since the update magnitude in GRPO is also governed by the within-group reward variance through $\hat{A}_i$, this analysis explains why DPE retains only moderately difficult samples to improve the learning efficiency per training example.

**Iterative training.** At iteration $k$, DPE first generates and validates a dataset $\mathcal{T}^{(k)}$ according to the diagnostic report, then applies difficulty-aware filtering to obtain $\mathcal{T}^{(k)}_{\text{train}}$, and finally performs GRPO to update the model: $\theta^{(k+1)} = \mathcal{A}_{\text{RL}}\left(\theta^{(k)}; \mathcal{T}^{(k)}_{\text{train}}\right)$. After the update, the system proceeds to the next diagnostic round and repeats the same procedure. This iterative process progressively strengthens weak capabilities and continuously expands visual coverage through external image sources, leading to a stable evolution of multimodal reasoning ability.

## 4. Experiments

### 4.1. Experiment Settings

**DPE Settings.** We evaluate DPE under extremely low-data conditions. Our framework uses only the first 1K samples from Vision-SR1-47K (Li et al., 2025a) as the seed dataset, whose initial category distribution is shown in Figure 4. Based on these 1K seed examples, the Multiple-Agents Questioner System generates approximately 4K training samples, while VisPlay uses 8K training samples per iteration. All other experimental configurations strictly follow those of VisPlay to ensure a fair comparison.

Due to the requirement for parallel data generation, the questioner system is implemented with 4 high-performance agents, including OpenAI o3 (OpenAI, 2025), Claude Sonnet 4 (Anthropic, 2025), Gemini-2.5-Pro (Comanici et al., 2025), and Qwen-VL-Max (Bai et al., 2025a). For image retrieval, we use the Serper API and retain the top 3 most relevant images in each search. For image editing and augmentation, we adopt Qwen-Image-Edit (Wu et al., 2025). The diagnostic mechanism is implemented using Qwen-VL-Max, which analyzes 200 randomly sampled problems from Vision-SR1-47K to identify the weaknesses of the target model.

**Baselines** We evaluate two base models, Qwen2.5-VL-7B-Instruct (Bai et al., 2025b) and Qwen3-VL-8B-Instruct (Bai et al., 2025a), and optimize them using either VisPlay or DPE. The number of evolution rounds is fixed to three for both methods following the VisPlay setting.

**Evaluation Protocol** We conduct evaluations using VLMEvalKit(Duan et al., 2024) and lmms-eval(Zhang et al., 2024a) on the following eleven benchmarks to ensure fair and reproducible results. STEM: MMMU (Yue et al., 2024a), MMVet (Yu et al., 2024), MMStar (Chen et al., 2024), and RealWorldQA (xAI, 2024). Visual Math: MathVerse (Zhang et al., 2024b), MathVision (Wang et al., 2024a), and MathVista (Lu et al., 2024). OCR: ChartQA (Masry et al., 2022) and CharXiv (Wang et al., 2024b). Multi-image: BLINK (Fu et al., 2024). Hallucination: HallusionBench (Guan et al., 2024). We use accuracy (Acc) as the evaluation metric.

### 4.2. Main Results

**Comparison with Self-evolving Method.** Table 1 demonstrates DPE's superiority over VisPlay across three key dimensions. First, comprehensive capability enhancement: DPE achieves consistent gains across STEM, OCR, and hallucination mitigation. On Qwen2.5-VL-7B-Instruct, it boosts CharXiv$_{\text{RQ}}$ by 4.11 points and outperforms VisPlay on HallusionBench (69.19% vs. 68.35%). Second, robust training dynamics: DPE mitigates the oscillation and regression observed in VisPlay. While VisPlay's performance on MMMU and BLINK fluctuates across iterations, DPE sustains

a smooth upward trend (e.g., MMMU 54.44 → 56.44), validating that our closed-loop mechanism effectively targets weaknesses without distribution drift. Finally, transferability: When applied to the stronger Qwen3-VL-8B-Instruct, DPE generalizes well, delivering substantial improvements on MMMU (+3.67) and MMStar (+10.86), proving its effectiveness across different model scales.

**Comparison with State-of-the-Arts.** As detailed in Table 2, DPE demonstrates remarkable parameter efficiency. Based on the 8B backbone, DPE achieves an average score of **64.39**, surpassing the 72B-parameter Qwen2.5-VL (61.9) and the proprietary GPT-4o (56.1). Notably, DPE dominates in complex reasoning tasks. In visual math, it establishes new SOTA performance on Math-Vista (**76.2**) and MathVision (**53.88**), significantly outperforming Qwen2.5-VL-72B by +1.4 and +15.7 points, respectively. In hallucination mitigation, DPE leads HallusionBench with **74.13**, demonstrating superior grounding compared to GPT-4o (67.5). These results suggest that data quality derived from DPE's closed-loop evolution is more critical than sheer parameter scale for solving complex multimodal problems.

### 4.3. Ablation Studies

**Impact of Static Data.** Table 3 highlights DPE's exceptional data efficiency. Despite using only ~3,000 iteratively generated samples—approximately $1/15$ of the static Vision-SR1-47K dataset—DPE achieves superior performance across key dimensions. Specifically, it improves MMMU (54.8 → 56.44), Hallusion-Bench (67.6 → 69.0), MathVista (68.8 → 69.5), and RealWorldQA (69.9 → 70.5). These results indicate that **the bottleneck in static training is not data volume, but rather the fixed distribution.** Static datasets inevitably lead to saturation on high-frequency patterns while neglecting long-tail capabilities, resulting in diminishing returns. In contrast, DPE's diagnostic module continuously identifies failure modes, allowing the generation process to concentrate the limited data budget on unresolved weaknesses. This closed-loop targeting overcomes the performance cap of fixed coverage, delivering broader and more stable improvements with substantially fewer samples.

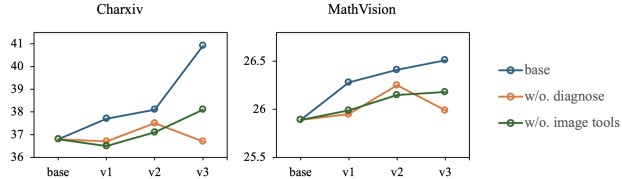

*Figure 3.* Ablation results on CharXiv and MathVision across three iterations, comparing full DPE with variants.

**Impact of the diagnostic module.** To assess the necessity of the diagnostic module, we remove it and repeat the three-iteration training procedure under the same settings. The results show that, without diagnostics, iterative gains become much smaller and noticeably less stable, and training tends to plateau or even regress. On CharXiv, full DPE achieves continuous improvements across iterations (36.8, 37.7, 38.1, 40.91), whereas removing diagnostics keeps performance close to the baseline (36.8, 36.7, 37.5, 36.7). In addition to yielding nearly zero net improvement, the ablated setting exhibits an improve then drop pattern, indicating that the training process can no longer reliably align with true capability gaps. A similar trend is observed on MathVision, where accuracy decreases from 26.25 at Iteration 2 to 25.99 at Iteration 3

*Table 1.* DPE's performance compared with Self-evolving Method's.

| Models | | Qwen2.5-VL-7B-Instruct | | | | | | | Qwen3-VL-8B-Instruct | | | |
|---|---|---|---|---|---|---|---|---|---|---|---|---|
| | | | VisPlay | | | DPE (Ours) | | | | DPE (Ours) | | |
| Methods | | Base | Iter 1 | Iter 2 | Iter 3 | Iter 1 | Iter 2 | Iter 3 | Base | Iter 1 | Iter 2 | Iter 3 |
| STEM | MMMU | 53.11 | 53.33 | 49.3 | 54.89 | 54.44 | 55.33 | **56.44** | 65.44 | 68.11 | 69.11 | **69.11** |
| | RealWorldQA | 68.63 | 68.50 | 68.10 | 69.02 | 69.41 | 69.54 | **70.46** | 71.63 | **71.63** | 70.72 | 70.85 |
| | MMVet | 67.20 | 67.84 | **69.44** | 67.52 | 67.71 | 67.02 | 68.35 | 67.29 | 70.92 | 70.00 | **72.80** |
| | MMStar | 63.27 | 63.60 | **66.07** | 65.07 | 65.00 | 64.60 | 65.60 | 61.27 | 71.40 | 71.67 | **72.13** |
| Visual Math | MathVerse | 43.12 | 42.97 | 42.71 | 44.19 | 43.78 | 44.26 | **45.10** | 53.22 | 55.99 | 56.47 | **57.18** |
| | MathVision | 25.89 | 25.00 | 26.18 | 25.72 | 26.28 | 26.41 | **26.51** | 51.97 | 52.04 | **55.03** | 53.88 |
| | MathVista | 65.50 | 64.80 | 68.8 | 68.20 | 67.50 | 68.20 | **69.50** | 76.20 | 76.60 | **78.00** | 76.20 |
| OCR | CharXiv$_{RQ}$ | 36.80 | 36.50 | 37.50 | 37.00 | 37.70 | 38.10 | **40.91** | 47.20 | 47.90 | 46.50 | **48.10** |
| | ChartQA | 85.64 | 86.04 | 86.04 | 86.16 | 86.12 | 86.16 | **86.56** | 85.08 | 84.84 | **85.20** | 84.80 |
| Specialized | HallusionBench | 64.98 | 64.98 | 68.35 | 68.24 | 69.09 | **69.19** | 68.98 | **74.24** | 73.92 | 74.13 | 74.13 |
| | BLINK | 56.02 | 56.44 | 56.55 | 55.65 | 56.18 | **56.65** | 56.23 | 68.54 | 68.96 | 68.12 | **69.22** |
| Overall | Average | 57.29 | 57.27 | 58.11 | 58.33 | 58.47 | 58.68 | **59.29** | 65.64 | 67.48 | 67.72 | **68.04** |

*Table 2.* DPE's Performance Compared with State-of-the-Arts'.

| Models | General Visual Understanding | | Visual Math | | | OCR | Hallucination | Avg |
|---|---|---|---|---|---|---|---|---|
| | MMMU | MMStar | MathVerse | MathVision | MathVista | CharXiv$_{RQ}$ | HallusionBench | |
| DeepEyes | - | - | 47.3 | 26.6 | 70.1 | - | - | - |
| DeepEyesV2 | - | - | 52.9 | 28.9 | 38.1 | 48.9 | - | - |
| Qwen2.5-VL-72B | 70.2 | 70.8 | **57.6** | 38.1 | 74.8 | 49.7 | 72.4 | 61.9 |
| GPT-4o | 69.1 | 64.7 | 50.2 | 30.4 | 63.8 | 47.1 | 67.5 | 56.1 |
| GPT5-Mini | 67.9 | 61.3 | 36.5 | 46.6 | 59.6 | 48.9 | 55.9 | 53.8 |
| Claude4-Sonnet | **75.1** | 67.4 | 65.9 | 52.7 | 72.4 | **60.9** | 54.5 | 64.1 |
| *Qwen3-VL-8B-Instruct* | | | | | | | | |
| DPE (Iter 3, Ours) | 69.11 | **72.13** | 57.18 | **53.88** | **76.2** | 48.1 | **74.13** | **64.39** |

without diagnostics, while full DPE steadily improves to 26.51. These results demonstrate that the diagnostic module is crucial for maintaining a correct and stable evolution direction. It significantly reduces distribution drift and performance oscillations commonly seen in self-evolving training, and ensures that each iteration continues to produce meaningful gains.

**Validating diagnosis-guided data distribution.** To further verify whether diagnostics truly guide the data distribution, Figure 4 visualizes the category distribution of the seed data and the mixture ratios recommended by the diagnostic module over three iterations. The diagnostic module does not simply follow the seed distribution or apply uniform sampling. Instead, it increases the sampling ratios of underperforming categories based on the failure patterns from the previous iteration, forming a targeted strengthening strategy driven by explicit error exposure.

More importantly, the redistribution aligns directly with the observed performance improvements. On CharXiv, the diagnostic module substantially increases the proportion of text-dense and chart-related samples in Iteration 1, and CharXiv accuracy immediately improves from 36.8 to 37.7, then further increases to 38.1 in subsequent iterations, demonstrating sustained and cumulative gains. In Iteration 2, the diagnostic module assigns more samples related to mathematical formulas and symbolic reasoning, and MathVision continues to improve (26.28, 26.41, 26.51), consistent with the upward trends on MathVerse and MathVista. These results indicate that the diagnostic module can effectively identify the model's current capability gaps and concentrate training resources on the truly weak dimensions through adaptive mixture

control, thereby improving the effectiveness of each iteration.

**Impact of image retrieval and editing.** We further analyze the contribution of the image retrieval and editing module (image tools) to iterative training. This module retrieves relevant samples from a larger external image pool and applies moderate editing and recomposition, which substantially expands visual diversity and long-tail coverage in the training data. Ablation results show that removing image tools makes the model more likely to reach an early plateau, and limits gains in later iterations, with particularly strong effects on OCR and chart-related tasks. As shown in Figure 3, on CharXiv, full DPE reaches 40.91 after three iterations, while removing image tools only reaches 38.1, resulting in a 2.81 drop. Moreover, most improvements occur in the first two iterations, with limited progress afterward. This suggests that generating text-level variations from the same or highly similar images can lead to overfitting to narrow layout and font distributions, failing to cover long-tail page structures and noise patterns, and thus capping performance on OCR tasks. A consistent effect is observed on MathVision, where removing image tools yields 26.18, lower than the 26.51 achieved by full DPE, indicating that visual diversity also improves robust perception of symbols, layouts, and localized regions in visual mathematical reasoning.

### 4.4. Diversity Analysis

We evaluate the diversity of generated data from both **textual** and **visual** perspectives. For a fair comparison, we embed questions and images using the same model, Qwen3-VL-Embedding. Let $f_{\text{text}}(\cdot)$ and $f_{\text{img}}(\cdot)$ denote its text and vision encoders. Given a

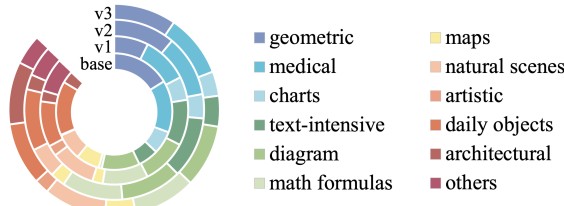

*Figure 4.* Category distribution of the seed set and the diagnosis-guided mixture ratios recommended by DPE over three iterations.

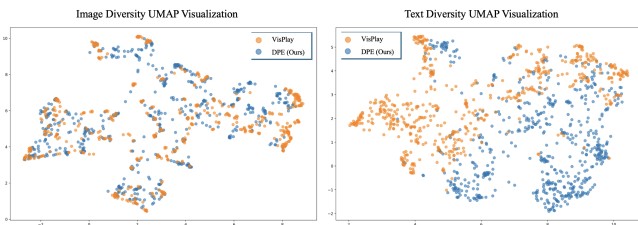

*Figure 5.* UMAP visualization of image diversity (left) and text diversity (right) for VisPlay and DPE.

question $q_i$ and image $I_i$, we obtain $z_i = f_{\text{text}}(q_i) \in \mathbb{R}^d$ and $v_i = f_{\text{img}}(I_i) \in \mathbb{R}^d$. For each iteration, we randomly sample $N = 200$ examples from VisPlay and DPE, respectively, and visualize their distributions under a shared UMAP projection.

**Text diversity.** We measure textual dispersion using the mean pairwise cosine distance:

$$\text{Diversity}(Z) = \frac{1}{N(N-1)} \sum_{i \neq j} \left(1 - \cos(z_i, z_j)\right). \quad (19)$$

As shown in Table 4, DPE achieves higher and more stable text diversity across three iterations. The base text diversity is 0.764. VisPlay increases to 0.83 at Iteration 1 but drops to 0.82 and 0.797 at Iterations 2 and 3. In contrast, DPE improves from 0.846 (Iter 1) to 0.866 (Iter 2) and remains high at 0.85 (Iter 3), indicating broader question coverage and reduced distribution collapse/template reversion. The UMAP in Fig. 5 further shows DPE covering a wider semantic region with additional subclusters.

**Visual diversity.** We apply the same metric to image embeddings and visualize them with UMAP. DPE also improves visual diversity over the base (0.835) to 0.847, 0.864, and 0.847 at Iterations 1–3. Since VisPlay mainly evolves from a fixed image set, Fig. 5 shows DPE covering a broader visual region with more non-overlapping content. This matches DPE's image retrieval and editing mechanism, indicating that it expands image sources and visual variations rather than only generating new text from static images, thereby improving long-tail visual coverage and maintaining diversity across iterations.

### 4.5. Quality Analysis of Generated Questions

We conduct a systematic quality evaluation of generated questions. At each iteration, we randomly sample 200 examples and ask three independent LLM judges (Claude Sonnet 4, OpenAI o3, and Gemini 2.5 Pro) to rate each example on a 5-point Likert scale from three aspects: **Clarity** (CL), **Solvability** (S), and **Correctness**

*Table 3.* Comparison between static training and DPE.

| Methods | Data Size | MMMU | HallusionBench | MathVista | RealWorldQA |
|---|---|---|---|---|---|
| *Qwen2.5-VL-7B-Instruct* | | | | | |
| Vision-R1 | 47K | 54.8 | 67.6 | 68.8 | 69.9 |
| DPE (Iter 3) | 3K | 56.44 | 69.0 | 69.5 | 70.5 |

*Table 4.* Text and image diversity scores (mean pairwise cosine distance) using Qwen3-VL-Embedding.

| Domain | Base | VisPlay | | | DPE (Ours) | | |
|---|---|---|---|---|---|---|---|
| | | Iter 1 | Iter 2 | Iter 3 | Iter 1 | Iter 2 | Iter 3 |
| Text | 0.764 | 0.830 | 0.820 | 0.797 | 0.846 | 0.866 | 0.850 |
| Image | 0.835 | 0.835 | 0.835 | 0.835 | **0.847** | **0.864** | **0.877** |

(CO). We report the overall quality score (QS) as the average across aspects and samples:

$$\text{QS} = \frac{1}{N} \sum_i \frac{CL_i + S_i + CO_i}{3}. \quad (20)$$

As shown in Table 5, DPE consistently produces higher-quality questions than VisPlay across all iterations and remains stable over time. VisPlay's QS is nearly unchanged from Iteration 1 to 2 (3.74→3.75) but drops to 3.32 at Iteration 3, indicating late-stage quality degradation. In contrast, DPE maintains near-ceiling QS values (4.96, 4.74, 4.80).

The improvements are mainly driven by *solvability* and *correctness*, i.e., whether questions are answerable from the image and whether answers are visually consistent. VisPlay's solvability declines to 2.98 at Iteration 3, while DPE stays above 4.86 (4.98, 4.86, 4.91). Similarly, VisPlay's correctness reaches 3.08 at Iteration 3, whereas DPE remains above 4.56 (4.93, 4.56, 4.58). Clarity shows the same trend: VisPlay drops from 4.38 to 3.91, while DPE stays close to 5 (4.99, 4.86, 4.92).

*Table 5.* Quality evaluation of generated questions.

| Metric | VisPlay | | | DPE (Ours) | | |
|---|---|---|---|---|---|---|
| | Iter 1 | Iter 2 | Iter 3 | Iter 1 | Iter 2 | Iter 3 |
| CL | 4.38 | 4.14 | 3.91 | 4.99 | 4.86 | 4.92 |
| S | 3.45 | 3.58 | 2.98 | 4.98 | 4.86 | 4.91 |
| CO | 3.40 | 3.52 | 3.08 | 4.93 | 4.56 | 4.58 |
| QS | 3.74 | 3.75 | 3.32 | 4.96 | 4.74 | 4.80 |

## 5. Conclusion

We introduce the Diagnostic-driven Pro- gressive Evolution (DPE) framework for Large Multimodal Models (LMMs) that overcomes heuristic self-evolution. By integrating a diagnostic-generation-reinforcement loop, our method explicitly identifies model blind spots, adaptively constructs targeted training data, and leverages large-scale unlabeled multimodal resources through cooperative multi-agent annotation. This design ensures controllable evolution directions, stable training dynamics, and sustained improvements on long-tail reasoning abilities that traditional self-expansion approaches fail to address. Experiments on several open-source LMMs show that our framework can deliver comprehensive reasoning enhancement using only a small amount

of training data, while detailed analyses further validate the crucial role of the diagnostic mechanism in improving stability and mitigating marginal utility saturation. Looking ahead, integrating richer diagnostic signals, expanding multimodal data sources, and exploring more sophisticated multi-agent collaboration strategies will continue advancing the development of adaptive, efficient, and continually improving multimodal reasoning systems.

## Impact Statement

This work aims to advance machine learning methodology by proposing a diagnostic-driven framework for improving multimodal reasoning in large models. The primary impacts of our approach are methodological: enabling more data-efficient capability enhancement, reducing training instability, and improving transparency in self-evolution pipelines. As with many techniques that enhance the reasoning abilities of large multimodal models, potential societal implications may include broader deployment of such systems in real-world applications. Risks related to synthetic data generation—such as the reinforcement of biases or propagation of inaccurate annotations—are mitigated through explicit validation mechanisms and controlled data distributions within our framework. Our experiments use only publicly available or synthetically generated inputs, thereby avoiding privacy concerns. Overall, we believe the ethical and societal considerations associated with this work are consistent with those commonly encountered in research on large-scale multimodal learning. No extraordinary risks beyond standard model improvements are introduced.

## Acknowledgment

This work was supported by the Shandong Province Taishan Scholars Young Experts Program under Grant No. tsqn202507081, and the Major Science and Technology Innovation Project of Shandong Province under Grant No. 2025CXGC010310.

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

## A. Case Study

Figure 6 presents qualitative comparisons. In Figure 6.(a), the question generated by VisPlay lacks necessary information and cannot be answered from the image content. In Figure 6.(b), VisPlay produces a multiple-choice question without options. In contrast, DPE generates questions with complete structure, sufficient information, and clear semantics, demonstrating its advantage in producing reliable training data.

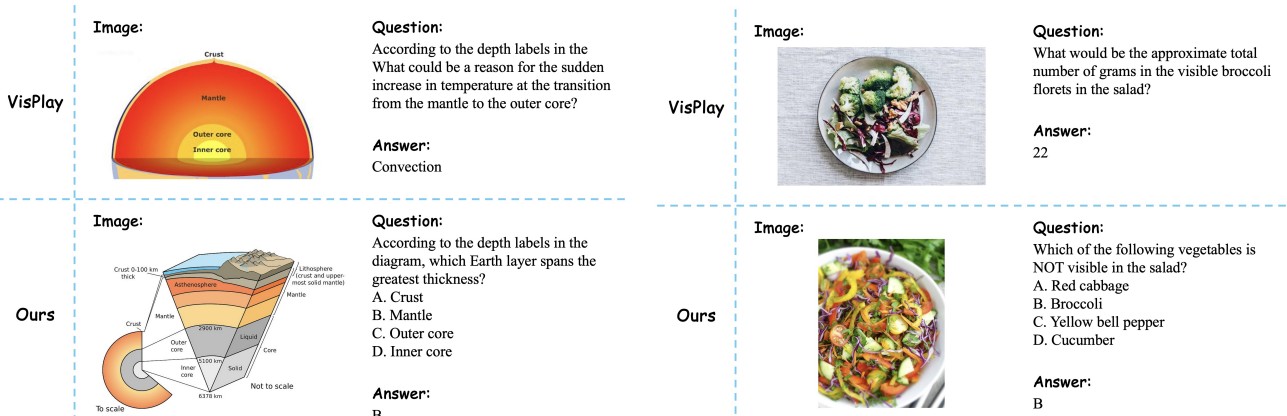

*(a)* VisPlay generates a question that lacks essential visual grounding and cannot be answered based on the image.

*(b)* VisPlay produces a multiple-choice question without options, resulting in an incomplete structure.

*Figure 6.* Case Study between VisPlay and our DPE framework.

