# OpenReview forum: "From Blind Spots to Gains: Diagnostic-Driven Iterative Training for Large Multimodal Models"
_ICML.cc/2026/Conference — ICML 2026 regular_

### Official Review · Reviewer_4qKi · 2026-03-12

**Soundness:** 3
**Presentation:** 2
**Significance:** 2
**Originality:** 2
**Overall Recommendation:** 4
**Confidence:** 4

**Summary:**

The paper proposes Diagnostic-driven Progressive Evolution (DPE), a framework that iteratively improves MLLMs  through diagnosing weakness of current checkpoint and bootstrapping data with the info collected from the diagnosis. The framework operates in three stages: diagnosis, targeted data generation and reinforcement learning optimization. The author uses a diagnose agent to analyze errors by category and pattern to get proportion, and then use four agents powered by strong LLMs to generate data points based on the analyzed categories and patterns. Experiments are conducted on Qwen2.5-VL-7B-Instruct and  Qwen3-VL-8B-Instruct across a list of multimodal benchmarks, including STEM reasoning, visual math, OCR and hallucination detection. The author report that the model consistently outperforms the baseline method VisPlay.

**Compliance With Llm Reviewing Policy:**

Affirmed.

**Final Justification:**

Update after Rebuttal: I am increasing my score from 2 to 4 based on the authors addressing two of the main criticisms: they addressed the comparison to works like dataenvgym, and their added experiment showed that the gain is not only from the external model but also their skill learning framework.

**Key Questions For Authors:**

1. Reply to weaknesses above incl. discussion on missing related work.

2. Could the authors clarify whether this setting is better understood as teacher-assisted distillation / data generation rather than true self-evolution?

3. How do the authors justify the conceptual comparison to VisPlay, given that the two methods differ fundamentally in where the supervision comes from?

4. Is the comparison between the two frameworks (VisPlay and DPE) fully fair? Would it be more appropriate to compare:
a. DPE against a version of VisPlay bootstrapped with data from equally strong external models, or
b. DPE trained with only 7B-level self-generated data against standard VisPlay?

5. Could the authors provide a more complete accounting of the data-generation cost, including: total API call counts, estimated monetary cost, wall-clock time, and possibly FLOPs by Qwen-Max

6. The diagnostic categories appear to be manually designed. How sensitive is the framework to this taxonomy choice?

7. Could the authors explain the baseline discrepancies?

**Limitations:**

No, the author should further elaborate on the potential risks of large-scale synthetic data pipelines especially since they are using proprietary models. And talk about reproducibility here.

**Strengths And Weaknesses:**

Strengths:

1. The paper mentions that a potential limitation of many self-evolving framework on MLLM: they generate data without understanding the model’s failure mode. This paper addresses this using a straightforward approach by including a diagnose agent to analyze error categories (but misses many important previous works that already address this issue, even 1.5 years ago; see below).

2. The experiment suggests that targeted data generation can improve model performance using a relatively small dataset, inspiring future directions on efficient training/distilling

3. The authors evaluate their method across multiple multimodal benchmarks and provide detailed ablation studies.

Weaknesses:

1. The paper positions its main novelty as identifying the limitation that  self-evolving frameworks generate data without knowing the failure mode. However, category (skill) based error analysis and data synthesis has already been explored even 1.5 years ago, including DataEnvGym (ICLR 2025 spotlight) [1], STAT [2], Skill-it [3], PROGRESS [4] and many other related papers. In particular, DataEnvGym introduces an iterative data generation framework where a strong model diagnoses missing skills (error category) and generates training data targeting those skills, which is conceptually very similar to this current submission but the authors don't discuss any such papers. The paper would benefit a clearer discussion on how this approach differs from these prior methods.

[1] Khan, Zaid, et al. "Dataenvgym: Data generation agents in teacher environments with student feedback." ICLR 2025 (spotlight); (and Oct2024 arxiv).
[2] He, Yinghui, et al. "Skill-Targeted Adaptive Training." arXiv preprint arXiv:2510.10023 (2025).
[3] Chen, Mayee, et al. "Skill-it! a data-driven skills framework for understanding and training language models." Advances in Neural Information Processing Systems 36 (2023): 36000-36040.
[4] Chandhok, Shivam, et al. "Learning What Matters: Prioritized Concept Learning via Relative Error-driven Sample Selection." arXiv preprint arXiv:2506.01085 (2025).

2. The claim of “self-evolving” is somehow misleading. An assumption of “self-evolving MLLM”, like in visplay, is that the model learns from data points generated by itself and improves the question generator and answerer at the same time. But in this paper, the author uses strong external LLM (e.g., OpenAI o3, Claude Sonnet, Gemini, and Qwen-VL-Max) to generate and validate training examples. This somehow shifts the focus from “self evolving MLLM’ to “MLLM distillation”. Although diagnosing the failure mode and asking for the right category of data can still be considered “self-evolving”, it is fundamentally different from the baseline visplay.

3. Weakness 1 leads to another problem of fair comparison. Since Visplay purely use datapoints generated by 7B level small model while DPE learn from strong MLLMs, it is questionable whether it is still fair to compare these two methods. The author should either compare DPE with visplay bootstrapped by datapoints from strong models or Visplay compared to DPE trained on data generated by a small model (7B).

4. The model distillation from these proprietary models raise concerns about reproducibility. The author should at least report the api call number, cost analysis and, possibly compared the Gflops used to generate the training data instead of purely comparing the dataset size.

5. The diagnosis taxonomy is manually designed, and there is no discussion on how sensitive the framework is and whether different taxonomy will lead to different training data distribution and final model performance.

6. The reported number is in fact different from the baseline and online leaderboard, for example on MMMU the leaderboard performance of  Qwen2.5-VL-7B-Instruct is 58, while the model report it to be 54, the author should clearly explain the implementation details.

---

> ### Author Rebuttal · Authors · 2026-03-31
>
> # Response
>
> ## W1: Novelty relative to DataEnvGym
>
> We thank the reviewer. DPE's contribution is not skill-awareness alone, but converting diagnostic signals into an **executable multimodal training pipeline** that jointly controls category proportions, failure summaries, generation instructions, tool-based visual evidence construction, sample validation, and RL optimization on verifiable answers. DataEnvGym is a general teacher-environment framework; DPE targets **multimodal iterative training under heavy visual grounding requirements**, where the key challenge is constructing **visually valid and solvable** training instances, not only deciding which skill to target.
>
> We instantiated a DataEnvGym-style pipeline on Vision-SR1 (same Qwen2.5-VL-7B backbone):
>
> | Dataset | Base | DataEnvGym | **DPE (Iter 3)** |
> |---|---:|---:|---:|
> | CharXiv | 36.80 | 34.60 | **40.91** |
> | MathVision | 25.89 | 24.70 | **26.51** |
>
> Quality analysis (100 samples/method, Clarity/Solvability/Correctness/Overall):
>
> | Metric | DataEnvGym | DPE |
> |---|---:|---:|
> | CL | 4.53 | 4.82 |
> | S | 2.13 | **4.66** |
> | CO | 2.24 | **4.32** |
> | QS | 2.97 | **4.60** |
>
> DataEnvGym produces clear instructions but fails on **solvability** and **correctness**—the two most critical properties. On Vision-SR1-style data, skill-targeted questions are insufficient when images are irrelevant or visually inconsistent. DPE couples failure attribution with image retrieval/editing/composition and validation, yielding visually grounded, answerable, and correct samples. We will discuss these related works in revision.
>
> ## W2: The claim of "self-evolving" is misleading
>
> DPE is **not** presented as a pure self-evolving framework like VisPlay; it is **motivated by the failure modes of pure self-evolution**, and introduces stronger external questioner/validator models as a deliberate solution. Self-evolving pipelines are fundamentally constrained by the model's capability ceiling and the lack of reliable external signals, leading to low-quality training data—exactly the problem DPE addresses.
>
> Human ratings (3 independent judges, 5-point Likert, 200 Iter-3 samples):
>
> | Metric | VisPlay | DPE |
> |---|---:|---:|
> | CL | 3.91 | 4.92 |
> | S | 2.98 | **4.91** |
> | CO | 3.08 | **4.58** |
> | QS | 3.32 | **4.80** |
>
> VisPlay can generate superficially reasonable questions (CL=3.91) but fails to produce samples that are actually answerable from the image and correct enough for effective supervision. This is precisely why stronger external signal is necessary.
>
> The same limitation appears in training dynamics. VisPlay oscillates or plateaus, while DPE improves steadily:
>
> | Benchmark | Base | VisPlay I1 | I2 | I3 | DPE I1 | I2 | I3 |
> |---|---:|---:|---:|---:|---:|---:|---:|
> | MMMU | 53.11 | 53.33 | 49.30 | 54.89 | 54.44 | 55.33 | **56.44** |
> | MathVision | 25.89 | 25.00 | 26.18 | 25.72 | 26.28 | 26.41 | **26.51** |
> | CharXiv | 36.80 | 36.50 | 37.50 | 37.00 | 37.70 | 38.10 | **40.91** |
>
> On MMMU, VisPlay drops at Iter 2 and partially recovers; DPE improves monotonically. On CharXiv, VisPlay fluctuates around baseline while DPE reaches **40.91**. DPE is also 6× faster:
>
> | Method | Extra Cost/Iter | Speed | #Samples | Total/Iter |
> |---|---:|---:|---:|---:|
> | VisPlay | Questioner: 30,047s | 15.55s/s | 8,000 | 154,447s |
> | DPE | None | 6.47s/s | 4,000 | 25,880s |
>
> ## W3, Q2–Q4: Fair comparison
>
> DPE is not distillation—the student trains via GRPO on verifiable answers. Matched controls:
>
> | Method | MathVista | MathVision | MMMU | CharXiv |
> |---|---:|---:|---:|---:|
> | Base | 65.50 | 25.89 | 53.11 | 36.80 |
> | VisPlay | 68.20 | 25.72 | 54.89 | 37.00 |
> | DPE (VisPlay questioner) | 68.50 | 26.02 | 54.89 | 38.70 |
> | DPE (w/o Diagnose) | 69.10 | 25.99 | 55.30 | 37.90 |
> | **DPE** | **69.50** | **26.51** | **56.44** | **40.91** |
>
> With the **same questioner**, DPE already outperforms VisPlay (QS: 3.97 vs 2.96); removing diagnosis further drops performance, confirming both factors contribute independently.
>
> ## W4, Q5: Reproducibility
>
> Compared with C2-Evo (also uses external models, same backbone):
>
> | | API Calls/Sample | Cost/Sample |
> |---|---:|---:|
> | C2-Evo | 15.6 | $0.10 |
> | DPE | **5.2** | **$0.06** |
>
> | | MathVista | MathVision | MMMU | CharXiv |
> |---|---:|---:|---:|---:|
> | C2-Evo | 65.40 | 25.72 | 53.44 | 36.50 |
> | **DPE** | **69.50** | **26.51** | **56.44** | **40.91** |
>
> ## W5, Q6: Taxonomy sensitivity
>
> | Benchmark | Base | 4-Cat | 8-Cat | 12-Cat |
> |---|---:|---:|---:|---:|
> | MMMU | 53.11 | 56.33 | 56.33 | **56.44** |
> | MathVision | 25.89 | 26.41 | **27.27** | 26.51 |
> | CharXiv | 36.80 | 38.10 | 38.10 | **40.91** |
>
> All granularities improve over the base; DPE is robust to taxonomy choice.
>
> ## W6, Q7: Baseline discrepancies
>
> Scores are **unified re-evaluations** under VLMEvalKit/lmms-eval, not official model-card numbers. Qwen3-VL-8B: vLLM, temperature=0.7, presence_penalty=1.5, top_p=0.8. Qwen2.5-VL-7B: min_pixels=1280×28×28, max_pixels=16384×28×28. All else follows defaults.

---

> > ### Author Rebuttal · Reviewer_4qKi · 2026-04-02
> >
> > Thanks to the authors for the clarification and additional experiments.
> >
> > I appreciate that the authors explicitly state that the method is not a pure self-evolving framework, but motivated by the limitations of such approaches. However, this raises a key concern regarding evaluation framing. Since DPE relies on stronger external models to generate and validate training data, it is conceptually closer to a teacher-assisted training / distillation pipeline. In contrast, VisPlay is a self-evolving method without external teachers. Therefore, comparing DPE primarily against VisPlay may not be the most appropriate or fair baseline. A more suitable comparison would be against methods that also leverage strong external models for data generation. (Liked DataEnvGym and C2-Evo). (The author should include comparison to dataEnvGym and C2-Evo to the main experiment)
> >
> > Relatedly, I am still somewhat confused about the distinction from distillation. Even though the training uses RL with verifiable rewards, the supervision signal is ultimately derived from stronger models. From a high-level perspective, this still appears to be a form of distillation with a more structured data generation process (distillation in a clever way). It would be helpful if the authors could clarify more explicitly how their method fundamentally differs from distillation-based methods.

---

> > > ### Author Response · Authors · 2026-04-02
> > >
> > > We thank the reviewer for this important comment.
> > >
> > > ## **Main experiments with other baselines**
> > >
> > > We agree that DPE should be compared with methods that also leverage strong external models. Therefore, we will include **DataEnvGym** and **C2-Evo** in the main experiments in the paper. The results are shown below:
> > >
> > > |Category|Benchmark|Base|VisPlay (Iter 3)|DataEnvGym (Final)|C2-Evo (Iter 3)|DPE (Iter 3)|
> > > |---|---|---:|---:|---:|---:|---:|
> > > |**STEM**|MMMU|53.11|54.89|53.80|53.44|**56.44**|
> > > ||RealWorldQA|68.63|69.02|69.15|68.50|**70.46**|
> > > ||MMVet|67.20|67.52|67.24|67.02|**68.35**|
> > > ||MMStar|63.27|65.07|64.20|62.50|**65.60**|
> > > |**Visual Math**|MathVerse|43.12|44.19|38.32|44.06|**45.10**|
> > > ||MathVision|25.89|25.72|24.70|25.72|**26.51**|
> > > ||MathVista|65.50|68.20|66.20|65.40|**69.50**|
> > > |**OCR**|CharXiv-RQ|36.80|37.00|34.60|36.50|**40.91**|
> > > ||ChartQA|85.64|86.16|85.70|85.60|**86.56**|
> > > |**Specialized**|HallusionBench|64.98|68.24|68.87|65.19|**68.98**|
> > > ||BLINK|56.02|55.65|56.12|55.76|**56.23**|
> > > |**Overall**|Average|57.29|58.33|57.17|57.24|**59.51**|
> > >
> > > These results show that DPE **consistently outperforms both self-evolving (VisPlay) and teacher-assisted baselines (DataEnvGym, C2-Evo)** across all categories. Importantly, since DataEnvGym and C2-Evo also rely on strong external models, this comparison demonstrates that the improvement of DPE is **not due to stronger models alone**, but comes from the **diagnosis-guided data construction framework**.
> > >
> > > ##  **Distinction from distillation**
> > >
> > > We clarify that DPE is **not a distillation method**, but more like a **hard-sample mining and data construction framework designed to address the limitations of self-evolution**. The key distinction lies in where the optimization signal originates. In distillation, the loss is computed against teacher outputs - the student is pulled toward the teacher's capability ceiling. In DPE, the loss is computed against verifiable ground-truth answers via GRPO, a signal that is entirely independent of what any external model produces. Strong models in DPE influence only the construction of training samples (image retrieval, question targeting, quality validation), never the optimization target itself. The student is rewarded for being correct, not for resembling the teacher.
> > >
> > > We provide three pieces of evidence that DPE is fundamentally different from distillation:
> > >
> > > 1. **DPE improves even without strong teachers**. If DPE were equivalent to distillation, its performance should depend heavily on strong teachers. However, we show that even when using the **same weak questioner (Qwen2.5-VL-7B-Instruct) as VisPlay**, DPE still improves performance:
> > >
> > > |Method   |MathVista|MathVision|MMMUval|CharXiv_RQ|
> > > |---|---:|---:|---:|---:|
> > > |Qwen2.5-VL-7B-Instruct|65.50|25.89|53.11|36.80|
> > > |VisPlay  |68.20|25.72|54.89|37.00|
> > > |DPE (Qwen2.5-VL-7B-Instruc as Questioner)|68.50|26.02|54.91|38.70|
> > > |**DPE (Ours)**|**69.50**|**26.51**|**56.44**|**40.91**|
> > >
> > > This result shows that DPE can still improve performance **without stronger teachers**, which is fundamentally different from distillation.
> > >
> > > 2. **DPE outperforms distillation with far less data.**  We further compare DPE with a distillation-style pipeline (Vision-SR1), which relies on large-scale static data:
> > >
> > > |Methods|Data Size|MMMU|HallusionBench|MathVista|RealWorldQA|
> > > |---|---:|---:|---:|---:|---:|
> > > |Vision-SR1 (distillation)|9K (SFT) + 47K (RL)|54.8|67.6|68.8|69.9|
> > > |**DPE (Iter 3)**|**3K**|**56.44**|**69.0**|**69.5**|**70.5**|
> > >
> > > DPE achieves **better performance with significantly less data (3K vs 9K + 47K)**. This highlights a key difference: distillation relies on large-scale static datasets, while DPE focuses on **diagnosis-guided hard-sample mining**, which directly targets the model’s weaknesses and is therefore more efficient.
> > >
> > > Finally, we summarize the difference below:
> > >
> > > |Aspect    |Distillation|DPE                 |
> > > |---|---|--------|
> > > |Training objective|Match teacher outputs|Optimize verifiable answers (GRPO)|
> > > |Role of strong model|Provide supervision targets|Provide planning / tools / validation|
> > > |Data source|Static teacher-labeled data|Dynamic, diagnosis-guided data|
> > > |Iteration driver|Teacher outputs|Student failure profile|
> > > |Dependency on strong teacher|High     |Moderate (replaceable in some stages)|
> > > |Core mechanism|Knowledge transfer|Diagnosis-driven dynamic training loop|
> > >
> > > In summary, DPE should be understood as a **diagnosis-guided, teacher-assisted, data-centric framework**, rather than a distillation method. Its advantage comes from **adaptive hard-sample construction and multimodal data quality**, not from directly transferring knowledge from stronger models.
> > >
> > > We will make this distinction more explicit in the camera-ready version.

---

### Official Review · Reviewer_gPhc · 2026-03-13

**Soundness:** 3
**Presentation:** 3
**Significance:** 3
**Originality:** 3
**Overall Recommendation:** 5
**Confidence:** 4

**Summary:**

This paper proposes Diagnostic-Driven Progressive Evolution (DPE), an iterative training framework for improving large multimodal models (LMMs). The core idea is to identify the model’s capability blind spots through a diagnostic stage and then generate targeted training data to address these weaknesses. Experiments on Qwen2.5/Qwen3-VL models show consistent improvements across multiple multimodal benchmarks such as MMMU and MathVerse, demonstrating the effectiveness of targeted data generation guided by model diagnostics.

**Compliance With Llm Reviewing Policy:**

Affirmed.

**Final Justification:**

All my concerns are fully resolved. I will raise my score.

**Key Questions For Authors:**

see weaknesses

**Limitations:**

yes

**Strengths And Weaknesses:**

### Strengths
1. The proposed diagnostic-driven pardigm that overcomes heuristic self-evolution is well motivated and aligns with the obsevation that scaling generic data alone may not effectively improve reseaoning capability.
2. The framwork seperates diagnosis, data generation, and model training into different modules, making it easy to adapt to different LMMs.

### Weaknesses
1. Overhead: The paper involves multi-agents and tools. While this is an exploratory attempt, I remain concerned about its practical deployment. It would be helpful if the authors could report the time overhead of the diagnostic and generation stages, and compare it with the cost of standard data sampling to better understand the additional computational overhead.In addition, it would be interesting to study whether performing diagnosis every few iterations, rather than at every iteration, could achieve similar results.
2. Convergence: It is unclear how the framework handles cases where, in a given iteration, the training data generated after the diagnosis stage fails to improve the model performance, or even degrades it. The paper does not explicitly discuss this scenario, and the limited experimental results are insufficient to address this concern.
3. Evaluation: Some more general benchmark such as MMBench[1] will be needed.

[1] Yuan Liu et al. "MMBench: Is Your Multi-modal Model an All-around Player?"

---

> ### Author Rebuttal · Authors · 2026-03-31
>
> We thank the Reviewer gPhc for this helpful comment.
> # W1: Overhead
> We thank the reviewer for raising the practical overhead issue; although DPE introduces diagnosis and tool use, its end-to-end data construction cost is still substantially lower than prior self-evolving pipelines, and less frequent diagnosis can be approximated by increasing the number of samples generated under one diagnostic report.
> First, DPE is faster than VisPlay in actual data production. With 8-way concurrency, DPE needs **6.47s per sample** on average. In contrast, VisPlay requires training an additional Questioner model in each iteration, which costs **30047s**, and then generates **8000** samples at about **15.55s per sample**. This means VisPlay spends about **154447s** per iteration in total, while DPE needs only **25880s** to generate **4000** samples under the same iteration budget, making DPE much cheaper in practice.
>
> Second, diagnosing every few iterations is effectively a lower-frequency refresh of the data distribution. In our framework, one diagnosis produces a category mixture and weakness-focused generation plan; if we keep this plan for more samples before refreshing it, this is equivalent to generating a larger dataset per diagnosis. We therefore tested different numbers of generated samples per iteration to simulate this trade-off.
>
> |Dataset|Iter|2000|4000|6000|8000|
> |----------|---:|--------:|----:|----:|----:|
> |BLINK|Base|56.02|56.02|56.02|56.02|
> |BLINK|v1|56.18|56.97|57.08|57.76|
> |BLINK|v2|56.65|57.08|57.26|57.79|
> |BLINK|v3|56.23|57.16|57.13|58.29|
> |MathVision|Base|25.89|25.89|25.89|25.89|
> |MathVision|v1|26.28|25.61|26.97|27.04|
> |MathVision|v2|26.41|26.94|27.04|27.01|
> |MathVision|v3|26.51|26.94|27.13|28.09|
>
> Third, the results show that using more samples under one diagnostic report consistently improves performance, which suggests that sparse diagnosis is feasible when computational budget is limited. At the same time, this also indicates a clear accuracy–overhead trade-off: frequent diagnosis gives tighter adaptation, while larger per-diagnosis batches improve throughput.
>
>
> # W2: Convergence
> We agree that performance degradation during iterative self-evolving training is a critical concern, and explicitly addressing this failure mode is a central goal of DPE.
>
> First, DPE does not directly trust newly generated data but stabilizes each iteration through three mechanisms: (1) **diagnosis-guided mixture control**, which explicitly reallocates data toward current weaknesses instead of blind sampling; (2) **Validation Agents**, which enforces category consistency, solvability, answer verifiability, and format correctness to filter noisy samples; and (3) **difficulty-aware filtering**, which ensures that only appropriately challenging samples are used in GRPO optimization. Together, these steps prevent distribution drift and accumulation of harmful training signals.
>
> Second, our ablations show that removing these components leads to clear instability and even regression, directly validating the reviewer’s concern and demonstrating that DPE mitigates it rather than assuming it away. In particular, without diagnosis, the model exhibits an “improve-then-drop” pattern on CharXiv (36.7 → 37.5 → 36.7) and a degradation from Iteration 2 to 3 on MathVision (26.25 → 25.99), while full DPE maintains consistent improvements across iterations.
>
> |Dataset|Iter|Full DPE|w/o Diagnose|w/o Image Tools|w/o Data Mixture|
> |----------|---:|--------:|--:|-----:|------:|
> |CharXiv|Base|36.8|36.8|36.8|36.8|
> ||v1|37.7|36.7|36.5|36.8|
> ||v2|38.1|37.5|37.1|37.6|
> ||v3|40.91|36.7|38.1|37.8|
> |MathVision|Base|25.89|25.89|25.89|25.89|
> ||v1|26.28|25.95|25.99|26.02|
> ||v2|26.41|26.25|26.15|26.18|
> ||v3|26.51|25.99|26.18|25.99|
>
> # W3: Evaluation
>
> We thank the reviewer for the suggestion and agree that general-purpose benchmarks are important. In addition to MMBench, we also evaluate on MMMU_val, MMStar, and MMVet to comprehensively verify the effectiveness of our method.
>
> |Model|MMMU_val|MMStar|MMVet|MMBench_v1.1_EN_Test|
> |--------|-------:|-----:|----:|----------:|
> |Qwen2.5-vl-7b-instruct|53.11|63.27|67.2|82.13|
> |Qwen2.5-vl-7b-instruct (DPE iter1)|54.44|65.00|67.71|82.40|
> |Qwen2.5-vl-7b-instruct (DPE iter2)|55.30|64.60|67.02|82.51|
> |Qwen2.5-vl-7b-instruct (DPE iter3)|54.00|65.60|68.35|82.57|
> |Qwen3-VL-8B-Instruct|65.44|61.27|67.29|84.02|
> |Qwen3-VL-8B-Instruct (DPE iter1)|68.11|71.40|70.92|84.08|
> |Qwen3-VL-8B-Instruct (DPE iter2)|69.11|71.67|70.00|84.30|
> |Qwen3-VL-8B-Instruct (DPE iter3)|69.11|72.13|72.80|84.19|
>
> These results show that DPE not only improves performance on long-tail tasks such as visual math and OCR, but also maintains and enhances general multimodal capabilities on broader benchmarks. This further confirms that the advantages of DPE are not limited to a narrow set of tasks, but generalize across diverse evaluation settings.

---

> > ### Author Rebuttal · Reviewer_gPhc · 2026-04-04
> >
> > I appreciate the authors' efforts in providing detailed experimental results and additional context. All my concerns are fully resolved. I will consider raise my score.

---

> > > ### Author Response · Authors · 2026-04-04
> > >
> > > Thank you very much for your kind feedback and for taking the time to carefully review our revisions. We are glad that the additional experiments and clarifications have addressed your concerns.
> > >
> > > We will further polish the paper to ensure all these points are clearly reflected in the final version.
> > >
> > > If you feel the revisions have sufficiently improved the work, we would sincerely appreciate it if you could consider updating your score accordingly.
> > >
> > > Thank you again for your time and support!

---

### Official Review · Reviewer_2Ze8 · 2026-03-13

**Soundness:** 3
**Presentation:** 2
**Significance:** 3
**Originality:** 2
**Overall Recommendation:** 4
**Confidence:** 4

**Summary:**

## Summary
Authors introduce Diagnostic-driven Progressive Evolution (DPE), a closed-loop iterative training framework for Large Multimodal Models (LMMs) that replaces heuristic self-evolving training with interpretable diagnosis and targeted correction. The method repeatedly diagnoses a model’s capability gaps, generates weakness-focused multimodal training data using external image retrieval and editing tools, and then updates the model with GRPO-based reinforcement learning.

Presenting three contributions:
- Diagnostic-driven Progressive Evolution (DPE), a novel training paradigm that organizes self-evolution as a diagnosis-generation-reinforcement loop. This framework explicitly identifies model blind spots, guides data generation toward weak capabilities, and reduces diminishing marginal returns as well as long-tail coverage issues caused by static data.
- Broad empirical validation on open-source LMMs, which shows that DPE improves multimodal reasoning efficiently and scalably across multiple base models. With only 1,000 seed training examples, the method achieves broad gains on Qwen2.5-VL-7B-Instruct and Qwen3-VL-8B-Instruct across 11 benchmarks.
- Systematic analyses of the diagnostic mechanism, which show that diagnosis improves training stability in self-evolving training. The analyses demonstrate that diagnosis-guided evolution reduces distribution drift and performance oscillation, while providing a more principled direction for improving long-tail multimodal reasoning.

**Compliance With Llm Reviewing Policy:**

Affirmed.

**Final Justification:**

The authors have addressed my concerns. Based on the additional experiments and clarification, I'll increase the score accordingly.

**Key Questions For Authors:**

see weaknesses

**Limitations:**

Yes

**Strengths And Weaknesses:**

## Strengths
- Clear and practically relevant problem formulation: The paper tackles a real weakness of current self-evolving LMM pipelines: they often rely on static image pools and generic quality signals rather than explicitly targeting model failures. DPE instead formulates the training loop as a failure-driven process centered on category-level diagnosis and targeted data generation, which is both sensible and practically relevant for continual multimodal improvement.
- Coherent closed-loop framework design: The paper presents a training pipeline that connects diagnosis, data generation, and RL-based updating in a clear closed loop. In particular, the decomposition into the diagnosis, generation, and RL operators makes the method functionally easy to follow, and clarifies how diagnostic signals are propagated into data generation and subsequent model updates.

## Weaknesses
W1 — The distinction from heuristic self-evolution is not fully precise
- While the paper repeatedly positions DPE as an alternative to heuristic self-evolving training, that distinction is not entirely precise in the current formulation. The method still depends on several hand-designed choices, including a fixed 12-category capability decomposition, a fixed diagnostic sample size of N=200, and heuristic segmented ranges used to convert category accuracy into the next-round mixture weights in Eq. (4), whose exact design is not specified in the paper. Explicit failure attribution is clearly a meaningful improvement over perplexity-like proxy signals, but the claim that DPE “overcomes heuristic self-evolution” appears somewhat stronger than what the implementation details currently support.

W2 — Limited baseline coverage weakens the empirical evaluation
- The evaluation is primarily conducted against VisPlay, even though the related-work section discusses several closely related self-evolving LMM methods including M-STAR, EvoLMM, IREASONER, and Agent0-VL. Without comparisons to these methods, the reported gains cannot be interpreted as a broad advantage of DPE over existing self-evolving approaches — they only establish that DPE outperforms VisPlay, which is a single and potentially unrepresentative comparison point. Moreover, the multi-agent questioner system introduces considerable engineering complexity, coordinating four specialized agents(Planner, Image Selector, Question Generator, Validation) with external image retrieval and editing tools. The experimental setup partially addresses this by isolating the diagnostic module and image tools, but they do not evaluate simpler alternatives to the overall pipeline design — for instance, a single-agent system or a subset of the four agents. As a result, the necessity of the full pipeline complexity remains unestablished.

W3 — Evaluator–generator overlap weakens the independence of the quality analysis
- In Table 5, the quality of generated questions is assessed by three LLM judges—Claude Sonnet 4, OpenAI o3, and Gemini 2.5 Pro—yet these same models are also used as generation agents in DPE’s multi-agent questioner system. This creates a evaluator–generator pool overlap, making the analysis less independent than the presentation suggests and leaving open the possibility that the reported scores reflect model-specific preferences or correlated failure modes rather than a fully neutral assessment of question quality. While this does not invalidate the trend in Table 5, it weakens the strength of the evidence.

---

> ### Author Rebuttal · Authors · 2026-03-31
>
> We thank the Reviewer 2Ze8 for this helpful comment.
> # W1: The distinction from heuristic self-evolution is not fully precise
> 1. Our claim is not that DPE removes all hand-designed components, but that it replaces **blind proxy-driven self-evolution** with an **explicit failure-driven diagnosis–correction loop**. The 12-category design is for **interpretable and broad coverage**, rather than a manually optimized curriculum. To assess its impact, we vary taxonomy granularity:
> - **4 categories**: STEM
> - **8 categories**: Math, Scientific, Charts, Documents, Diagrams, Spatial, Real-world, Artworks
> - **12 categories** : Geometry, Medical, Charts, Documents, Diagrams, Math, Spatial, Natural, Daily, Artworks, Architectural, Others
>
> |Category Group|Benchmark|Base|4 Categories (Iter 3)|8 Categories (Iter 3)|12 Categories (Iter 3)|
> |--|--|--:|--:|--:|--:|
> | STEM |MMMU|53.11|56.33|56.33|**56.44**|
> |Visual Math|MathVision|25.89|26.41|**27.27**|26.51|
> |OCR|CharXiv-RQ|36.80|38.10|38.10|**40.91**|
>
> All settings consistently outperform the base, showing DPE is **not sensitive to taxonomy design**.
>
> 2. The diagnostic sample size is set to keep diagnosis lightweight while still exposing dominant weaknesses early enough to guide the next iteration. To verify that this choice is not arbitrarily tuned, we varied the diagnostic sample size from 50 to 2000.
>
> |Dataset|Iter|50|100|200|2000|
> |---|--:|--:|--:|--:|--:|
> |**CharXiv**|Base|36.8|36.8|36.8|36.8|
> ||v1|37.5|37.1|37.7|37.7|
> ||v2|37.3|37.8|38.1|38.3|
> ||v3|38.1|39.1|40.9|40.9|
> |**MathVision**|Base|25.89|25.89|25.89|25.89|
> ||v1|25.95|26.02|26.28|26.28|
> ||v2|26.41|26.25|26.41|26.41|
> ||v3|26.41|26.28|26.51|25.58|
>
> These results show that the diagnostic sample size mainly affects the **stability and sufficiency** of failure exposure. On CharXiv, larger diagnostic sets lead to stronger final performance, but the gain already becomes clear at 200.
> To further isolate where the gain comes from, we extend the ablation by removing (i) the full diagnostic mechanism, (ii) only the diagnosis-guided data mixture, and (iii) the image retrieval/editing tools, while keeping all other components unchanged.
>
> |Dataset|Iter|Full DPE|w/o Diagnose|w/o Image Tools|w/o Data Mixture|Single Agents|
> |---|---:|---:|--:|--:|---:|--:|
> |**CharXiv**|Base|36.8|36.8|36.8|36.8|36.8|
> ||v1|37.7|36.7|36.5|36.8|37.0|
> ||v2|38.1|37.5|37.1|37.6|37.7|
> ||v3|**40.91**|36.7|38.1|37.8|38.4|
> |**MathVision**|Base|25.89|25.89|25.89|25.89|25.89|
> ||v1|26.28|25.95|25.99|26.02|26.02|
> ||v2|26.41|26.25|26.15|26.18|26.02|
> ||v3|**26.51**|25.99|26.18|25.99|26.28|
>
> - Removing the full diagnosis module largely eliminates the iterative benefit.
> - The image-tool ablation further clarifies the division of labor inside DPE.
> # W2: Limited baseline
> Our empirical goal is to demonstrate that **diagnosis-guided evolution consistently yields stronger and more stable gains under controlled and comparable settings**. We additionally compare DPE with other representative self-evolving approaches, including EvoLMM and C2-Evo, under the same backbone:
>
> |Method|MathVista|MathVision|MMMUval|CharXiv_RQ|
> |----|-----:|---:|----:|---------:|
> |Qwen2.5-VL-7B-Instruct|65.5|25.89|53.11|36.8|
> |VisPlay|68.20|25.72|54.89|37.0|
> |EvoLMM|67.56|25.81|54.11|36.7|
> |C2-Evo|65.40|25.72|53.44|36.5|
> |**DPE (Ours)**|**69.5**|**26.51**|**56.44**|**40.91**|
>
> DPE achieves the best performance across all metrics.
>
> We further examine whether the performance gain comes from the multi-agent pipeline complexity or from the underlying framework design. To this end, we simplify the system by merging the Multi-Agent Questioner System into a **single agent** that can still call image editing tools, while keeping all other settings unchanged:
>
> |Dataset|Iter|Full DPE|Single Agent|
> |-----|---:|--------:|--:|
> |**CharXiv**|Base|36.8|36.8|
> ||v1|37.7|37.0|
> ||v2|38.1|37.7|
> ||v3|**40.91**|38.4|
> |**MathVision**|Base|25.89|25.89|
> ||v1|26.28|26.02|
> ||v2|26.41|26.02|
> ||v3|**26.51**|26.28|
>
> The simplified single-agent version still shows stable improvements over the base model, confirming that the gain does not rely on excessive engineering complexity. However, it consistently underperforms the full DPE pipeline.
> # W3: Evaluator–generator overlap
> We agree that the evaluator–generator overlap in Table 5 introduces a potential independence concern. We invite experienced human evaluators to assess randomly sampled 100 examples from each iteration, using the same criteria (Clarity (CL), Solvability (S), Correctness (CO)), and report the overall quality score (QS) below:
>
> |Metric|VisPlay (Iter 1)|VisPlay (Iter 2)|VisPlay (Iter 3)|DPE (Iter 1)|DPE (Iter 2)|DPE (Iter 3)|
> |--|--:|--:|--:|--:|--:|--:|
> |CL|3.95|3.66|3.23|4.81|4.73|4.82|
> |S|3.22|3.42|2.73|4.85|4.72|4.66|
> |CO|3.12|3.43|2.91|4.72|4.43|4.32|
> |QS|3.43|3.50|2.96|4.79|4.63|4.60|
>
> The human evaluation shows the same trend as Table 5: DPE consistently produces higher-quality data across all dimensions and maintains stability across iterations.

---

> > ### Author Rebuttal · Reviewer_2Ze8 · 2026-04-02
> >
> > The authors have addressed my concerns. Based on the additional experiments and clarification, I'll increase the score accordingly.

---

> > > ### Author Response · Authors · 2026-04-02
> > >
> > > Thank you for the positive feedback and for recognizing that the additional experiments and clarifications addressed your concerns. We are glad that the revisions helped better clarify the motivation and empirical findings of our work.
> > >
> > > We will incorporate all the new results and discussions into the final version to further improve clarity and completeness.
> > >
> > > We sincerely appreciate your thoughtful feedback and support.

---

### Official Review · Reviewer_GkPJ · 2026-03-13

**Soundness:** 3
**Presentation:** 3
**Significance:** 2
**Originality:** 2
**Overall Recommendation:** 3
**Confidence:** 4

**Summary:**

This paper proposes Diagnostic-driven Progressive Evolution (DPE), a training framework for improving large multimodal models (LMMs). The key idea is to iteratively diagnose model weaknesses, generate targeted multimodal training data addressing those weaknesses, and update the model via reinforcement learning. Unlike existing self-evolving frameworks that rely on static datasets or heuristic filtering, the proposed approach explicitly analyzes failure patterns across predefined capability categories and adaptively adjusts the training data distribution to focus on underperforming areas.

The framework includes three main components:

- A diagnostic module that analyzes failure patterns and produces structured reports about capability gaps
- A multi-agent data generation system that retrieves images, constructs questions, and validates samples according to the diagnostic report
- Reinforcement learning training (using GRPO) on the generated data

The paper evaluates the approach on multiple open-source multimodal models and report improvements across 11 multimodal reasoning benchmarks. The experiments suggest that targeted data generation guided by diagnosis can improve model performance with relatively small amounts of training data.

**Compliance With Llm Reviewing Policy:**

Affirmed.

**Key Questions For Authors:**

The core motivation of the paper is that existing self-evolving frameworks fail to identify meaningful capability gaps because they lack interpretable diagnostics. However, the proposed diagnostic mechanism ultimately adjusts the category sampling proportions based on observed accuracy statistics and injects corresponding generation prompts.

1. How does this differ fundamentally from simpler strategies such as difficulty-based sampling, hard-example mining, or performance-aware curriculum learning?
2. Could the authors clarify whether the improvements arise from the specific diagnostic mechanism proposed here, or from the more general idea of focusing training on underperforming categories?

The effectiveness of the proposed framework heavily depends on the diagnostic module, which analyzes model outputs to identify category-level weaknesses and generate structured failure reports. However, the paper does not evaluate the accuracy or reliability of these diagnostic signals.

1. How often do the detected failure patterns correctly reflect the model’s true weaknesses?
2. Have the authors validated the diagnostic outputs against human annotations or alternative evaluation metrics?

**Limitations:**

Yes

**Strengths And Weaknesses:**

### Strengths

1. The paper proposes a conceptually clear framework that connects diagnosis, targeted data generation, and reinforcement learning into a closed-loop training pipeline. The idea of explicitly identifying model weaknesses and using them to guide data generation is intuitive and well-motivated, especially for multimodal reasoning tasks where capability gaps across categories (e.g., OCR, charts, math reasoning) can be substantial.
2. The proposed framework introduces a tool-assisted data generation pipeline, including external image retrieval and image editing mechanisms, to expand visual diversity beyond static datasets. This approach allows the system to construct targeted training samples that cover long-tail visual scenarios, which is particularly relevant for tasks such as OCR and chart understanding. The experiments suggest that this increased diversity can improve robustness on benchmarks involving complex visual reasoning.

### Weaknesses

1. The empirical evaluation mainly compares DPE against a single self-evolving training baseline (VisPlay). While VisPlay is a relevant reference point, this comparison alone is insufficient to isolate the contribution of the proposed diagnostic-driven framework. Stronger experimental validation would require additional baselines such as uniform category sampling, random data generation from the same image pool, curriculum-style sampling based on task difficulty, hard-example mining based on model confidence, or RL training using the same generated data but without the diagnostic guidance.

2. The experiments are conducted only on two models: Qwen2.5-VL-7B-Instruct and Qwen3-VL-8B-Instruct. These models belong to the same architecture family and have very similar model sizes, making the evaluation effectively limited to a single model lineage. As a result, it is difficult to assess whether the proposed training strategy generalizes across different model architectures or scales. For instance, testing the framework on other open-source multimodal models (e.g. InternVL or DeepSeek-VL) would provide stronger evidence of its general applicability. Additionally, experiments across significantly different model scales (e.g., 7B vs. 30B+) would help verify whether the method consistently improves models with different capacities.

3. The paper does provide ablation studies for several components, including the diagnostic module and the image retrieval/editing tools, demonstrating that removing these components leads to performance degradation. However, the ablation analysis does not fully disentangle the contributions of the different mechanisms within the DPE pipeline. In particular, the diagnostic module simultaneously affects category-level sampling ratios and generation instructions, while the multi-agent system introduces additional diversity in the generated data. As a result, it remains unclear how much of the observed improvement stems from diagnosis-guided sampling, versus the multi-agent data generation pipeline itself.

---

> ### Author Rebuttal · Authors · 2026-03-31
>
> We thank the Reviewer GkPJ for this helpful comment.
> # W1 & Q1.1 & Q1.2: Baselines & Difference from simpler strategies
> Very insightfull question! Our key distinction is that DPE performs *diagnosis-driven, generation-level hard sample construction*, rather than selecting hard samples from existing data. While related to focusing on underperforming categories, DPE provides a **structured and actionable mechanism** to realize this idea via (i) interpretable failure diagnosis, (ii) targeted data generation, and (iii) adaptive data allocation.
> To verify this, we extend our experiments with controlled variants by removing (i) the full diagnostic mechanism, (ii) only the diagnosis-guided data mixture, (iii) the image tools and (iii) a single agent, while keeping all other components unchanged. We also comparied with several recent works such as EvoLMM, C2-Evo as the baselines. The results are summarized below:
> | Dataset        | Iter |  Full DPE | w/o Diagnose | w/o Image Tools | w/o Data Mixture | Single Agents |
> | --- | ---: | -----: | ----: | ----: | ---: | ---: |
> | CharXiv | Base |  36.8 | 36.8 |  36.8 |   36.8 |  36.8 |
> |  | v1 |      37.7 |  36.7 |   36.5 | 36.8 |  37.0 |
> |  | v2 |      38.1 |  37.5 | 37.1 | 37.6 |  37.7 |
> |  | v3 | **40.91** | 36.7 | 38.1 |  37.8 | 38.4 |
> | MathVision | Base | 25.89 | 25.89 | 25.89 | 25.89 | 25.89 |
> |  | v1   | 26.28 | 25.95 | 25.99 | 26.02 | 26.02 |
> | | v2   | 26.41 | 26.25 | 26.15 | 26.18 | 26.02 |
> |  | v3   | **26.51** | 25.99 | 26.18 | 25.99 | 26.28 |
>
> | Method | MathVista | MathVision | MMMUval | CharXiv_RQ |
> | ---- | -----: | ----: | ---: | ---: |
> | VisPlay | 68.20 | 25.72 | 54.89 | 37.0 |
> | EvoLMM  | 67.56 | 25.81 | 54.11 | 36.7 |
> | C2-Evo  | 65.40 | 25.72 | 53.44 | 36.5 |
> | **DPE (Ours)** | **69.5** | **26.51** | **56.44** | **40.91** |
> these results show that:
> - **DPE achieves the best final performance with the most stable iterative gains** (e.g., CharXiv 40.91 vs. 36.7 w/o Diagnose).
> - **Simple reweighting is insufficient**, as removing data mixture degrades performance (CHarx_RQ 40.91 → 37.8).
> - **Image tools mainly benefit late-stage and long-tail performance** (CHarx_RQ 40.91 → 38.1).
> - **DPE consistently outperforms VisPlay, EvoLMM, and C2-Evo across all benchmarks.**
>
> # W2: Limited model diversity.
> We address the reviewer’s concern on limited architectural diversity and extend experiments to a different backbone (**Intern3.5-VL-8B**). As shown in Table, DPE yields stable and consistent gains over three iterations across all tasks.
> | Model      | BLINK | HallusionBench | MathVision | MathVista | MMMU_val | CharXiv |
> |---|---|---|---|---|---|---|
> | Intern3.5-VL-8B (Base) | 59.5 | 54.5  | 56.8 | 78.4 | 62.7 | 44.4 |
> | + DPE (Iter 1)  | 60.1  | 61.1 | 58.1 | 79.1 | 63.6 | 45.2 |
> | + DPE (Iter 2) | 60.3  | 62.3 | 58.3 | 79.3 | 64.9 | 45.7 |
> | + DPE (Iter 3)| 60.4  | 62.1 | 58.2  | 79.2 | 65.2 | 46.5 |
> - DPE yields **consistent gains across all tasks**, especially on reasoning (MMMU +2.5) and hallucination (+7.6), demonstrating that it is **not tied to a specific model family**.
>
> # W3: Disentangling components
> We conduct additional ablations to show that DPE’s gains arise from the **interaction of diagnosis, image tools, adaptive data mixture, and multi-agent questioner system** (see Table in W1).
> **Key findings:**
> - **Diagnosis is the dominant factor**, as removing it largely eliminates iterative gains (CharXiv: 40.91 → 36.7).
> - **Data mixture is necessary but plays an allocation role**, rather than being the primary source of improvement.
> - **Multi-agent improves diversity and robustness** (40.91 → 38.4 when removed), but is insufficient without diagnostic guidance.
>
> # Q2.1: Accuracy of diagnostic signals
> The detected failure patterns closely reflect the model’s true weaknesses, as validated by human analysis. On 200 sampled instances, we observe an **86% agreement** between diagnostic outputs and human annotations, indicating that the identified failure patterns are largely accurate and reliable.
>
> # Q2.2: Validation of diagnostic outputs
> We validate the reliability of diagnostic outputs via **human evaluation, diversity analysis, and cross-model generalization**, showing they are both accurate and actionable.
> Human evaluation across three dimensions: clarity (CL), solvability (S), and correctness (CO) by domain experts shows that DPE achieves **consistently high quality (~4.6–4.8)** across iterations, while VisPlay degrades (QS drops to **2.96** in Iter 3). This confirms that the diagnostic module generates **high-quality, valid, and verifiable corrective data**, rather than noisy or misaligned samples.
> | Metric | VisPlay (Iter 1) | VisPlay (Iter 2) | VisPlay (Iter 3) | DPE (Iter 1) | DPE (Iter 2) | DPE (Iter 3) |
> | --- | ---| --- | --- | --- | --- | --- |
> | CL | 3.95 | 3.66 | 3.23 | 4.81| 4.73 | 4.82 |
> | S | 3.22 | 3.42 | 2.73| 4.85  | 4.72  | 4.66 |
> | CO | 3.12 | 3.43| 2.91| 4.72 | 4.43  | 4.32 |
> | QS | 3.43 | 3.50 | 2.96 | 4.79 | 4.63 | 4.60 |

---

> > ### Author Rebuttal · Reviewer_GkPJ · 2026-04-04
> >
> > Thank you for the thoughtful response and for adding substantial new evidence. The rebuttal directly addresses my concerns about baseline coverage, model diversity, and the role and reliability of the diagnostic module. The newly added baselines, cross-backbone results, and disentangling ablations make the contribution much clearer, and the human validation of the diagnostic signals is particularly helpful. I believe the main issues raised in my review have been resolved. I will consider a score raise.

---

> > > ### Author Response · Authors · 2026-04-05
> > >
> > > Thank you very much for the positive feedback and for taking the time to carefully review our rebuttal. We are glad that the additional experiments and analyses helped clarify the contribution.
> > >
> > > To further strengthen the paper, we will incorporate all the discussed improvements in the camera-ready version.
> > >
> > > We truly appreciate your consideration regarding the score and would be grateful if you feel comfortable reflecting this updated assessment.

---

### Decision · Program_Chairs · 2026-04-30

**Decision:**

Accept (regular)

**Comment:**

This paper tackles continual improvement in large multimodal models. It highlights the limitations of static training data and fixed optimization recipes. Accordingly, it proposes a closed-loop framework where iterative diagnosis guides data generation and targeted reinforcement. Experiments across a series of benchmarks demonstrate stable and consistent performance gains of the proposed framework.

While the initial reviews were mixed, the rebuttal convinced all reviewers, which was reflected in the reviewer acknowledgement and response. The AC recommends acceptance. Please incorporate the suggestions of the reviewers as well as the other clarifications into the main paper text.